# The Hausa Back Beliefs Questionnaire: Translation, cross-cultural adaptation and psychometric assessment in mixed urban and rural Nigerian populations with chronic low back pain

Aminu Alhassan Ibrahim[1,2]*, Mukadas Oyeniran Akindele[1], Sokunbi Oluwaleke Ganiyu[1], Bashir Kaka[1], Bashir Bello[1]

1 Faculty of Allied Health Sciences, Department of Physiotherapy, College of Health Sciences, Bayero University Kano, Kano, Kano State, Nigeria, 2 Department of Physiotherapy, Muhammad Abdullahi Wase Teaching Hospital, Hospitals Management Board, Kano, Kano State, Nigeria

* amenconafs@gmail.com

**Data Availability Statement:** All relevant data are within the manuscript and its Supporting Information files.

## Abstract

### Introduction

Negative attitudes and beliefs about low back pain (LBP) can lead to reduced function and activity and consequently disability. One self-report measure that can be used to assess these negative attitudes and beliefs and to determine their predictive nature is the Back Beliefs Questionnaire (BBQ). This study aimed to translate and cross-culturally adapt the BBQ into Hausa and assess its psychometric properties in mixed urban and rural Nigerian populations with chronic LBP.

### Methods

The BBQ was translated and cross-culturally adapted into Hausa (Hausa-BBQ) according to established guidelines. To assess psychometric properties, a consecutive sample of 200 patients with chronic LBP recruited from urban and rural clinics of Nigeria completed the questionnaire along with measures of fear-avoidance beliefs, pain catastrophizing, functional disability, physical and mental health, and pain. One hundred of the 200 patients completed the questionnaire twice at an interval of 7–14 days to assess test-retest reliability. Internal construct validity was assessed using exploratory factor analysis, and external construct validity was assessed by examining convergent, divergent, and known-groups validity. Reliability was assessed by calculating internal consistency (Cronbach's α), intraclass correlation coefficients (ICC), standard error of measurement (SEM), minimal detectable change at 95% confidence interval ($MDC_{95}$), and limits of agreement using Bland-Altman plots. Reliability (ICC, SEM and $MDC_{95}$) was also assessed separately for rural and urban subgroups.

**Funding:** This study received no specific grant from any funding agency in the public, commercial, or not-for-profit sectors.

**Competing interests:** The authors declare no competing interests.

## Results

The factor analysis revealed a four-factor solution explaining 58.9% of the total variance with the first factor explaining 27.1%. The nine scoring items loaded on the first factor hence supporting a unidimensional scale. The convergent and divergent validity were supported as 85% (6:7) of the predefined hypotheses were confirmed. Known-groups comparison showed that the questionnaire discriminated well for those who differed in education ($p < 0.05$), but not in age ($p > 0.05$). The internal consistency and ICC (α = 0.79; ICC = 0.91) were adequate, with minimal SEM and $MDC_{95}$ (1.9 and 5.2, respectively). The limits of agreements were –5.11 to 5.71. The ICC, SEM and $MDC_{95}$ for the urban and rural sub-groups were comparable to those obtained for the overall population.

## Conclusions

The Hausa-BBQ was successfully adapted and psychometrically sound in terms of internal and external construct validity, internal consistency, and test-retest reliability in mixed urban and rural Hausa-speaking populations with chronic LBP. The questionnaire can be used to detect and categorize specific attitudes and beliefs about back pain in Hausa culture to prevent or reduce potential disability due to LBP.

## Introduction

Low back pain (LBP) is a common musculoskeletal disorder and presently the leading cause of disability in both developed and developing countries [1]. Nearly all individuals will, at some point in their life, experience LBP [2]. Although most episodes of LBP are benign in nature, some fraction of individuals may develop recurrent or chronic pain, which is accountable for the majority of direct and indirect costs associated with LBP [3]. Thus, chronic LBP is an important public health problem in the world necessitating attention in research, and effective health care [4].

Contemporary understanding suggests that LBP is a complex disorder associated with multiple contributors to both pain and related disability, including biophysical factors, psychosocial factors, comorbidities, and pain-processing mechanisms [5,6]. Specifically, psychosocial factors have been well documented to have a significant impact on pain persistence and the development of chronic disability [7–11]. Moreover, the impact of psychosocial factors does not only include pain experience but also treatment outcomes and consequently recovery [12,13].

One important modifiable psychological factor related to LBP disability is back pain beliefs. According to Vlaeyen et al. [14], negative beliefs about back pain, are often viewed as a signal of an impending threat, which leads to fear of movement/(re)injury, decreased function and activity, and consequently persistent chronic disability. In support of this notion, results of several cross-sectional studies suggest that negative beliefs about back pain are associated with persistent, high levels of pain and disability [15–18], care-seeking behavior [19], work absenteeism, and reduced productivity [20–22]. Additionally, studies have shown that back pain beliefs are influenced by culture [23] and demographic characteristics such as age, education level, and working environment [16,24,25].

The Back Beliefs Questionnaire (BBQ) developed by Symonds et al. in 1996 [20] is a widely used self-reported outcome to assess attitudes and beliefs towards recovery and return-to-

work. The questionnaire has been used as an outcome to test the effectiveness of interventions targeting back pain beliefs [26–28]. Furthermore, it has been shown to predict recovery rate from LBP [29]. The original English BBQ proved to be valid and reliable [20,30], and has been adapted and validated for use in several languages/cultures [31–41].

The three main indigenous languages in Nigeria are Hausa, Igbo, and Yoruba. Recently, the BBQ was successfully adapted into Yoruba [42]. However, no Hausa version of this tool is available despite Hausa being the largest ethnic group not only in Nigeria but also in sub-Saharan Africa with about 80 million speakers [43]. The language is commonly spoken in Benin, Cameroon, Chad, Central African Republic, Eritrea, Equatorial Guinea, Gabon, Gambia, Ghana, Ivory Coast, Niger, Republic of the Congo, Senegal, Sudan, and Togo [44]. Adapting the BBQ into Hausa will enhance the uptake of studies in these regions as it may assist researchers and clinicians to detect negative attitudes and beliefs about back pain and design appropriate interventions. Therefore, this study aimed to translate and cross-culturally adapt the BBQ into Hausa and assess its psychometric properties in mixed urban and rural Nigerian populations with chronic LBP.

## Methods

### Ethical considerations

This study was approved by the Health Research Ethics Committee, Kano State Ministry of Health, Nigeria (Ref: MOH/Off/797/T.I./651). Written permission to translate the English version of the BBQ into Hausa was obtained from the original developers. Written informed consent was obtained from all participants before their involvement in the study.

### Study design

This cross-sectional study was conducted in two stages: Translation and cross-cultural adaptation of the BBQ into Hausa; and assessment of psychometric properties of the translated version.

### Outcome measures

**Back Beliefs Questionnaire (BBQ).**    The BBQ is a 14-item scale, with each item rated using a 5-point Likert scale ranging from strongly disagree (1) to strongly agree (5). Five of the items (4, 5, 7, 9, and 11) are distractors and nine items (1, 2, 3, 6, 8, 10, 12, 13, and 14) are used for scoring of the questionnaire resulting in a total score ranging from 9 to 45 [20]. The score obtained for each item is reversed (for example 5 means 1, and 2 means 4) before summing to obtain the final score meaning the lower the scores, the more pessimistic beliefs regarding the consequences of back pain [20]. The original English BBQ demonstrated excellent internal consistency (Cronbach's α: 0.70) and test-retest reliability (intraclass correlation coefficients [ICC]: 0.87).

**Hausa Fear-avoidance Beliefs Questionnaire (FABQ).**    The FABQ assesses fear-avoidance beliefs about physical activity and work [45]. It consists of 16 items, with each item rated using a Likert scale ranging from 0 (completely disagree) to 6 (completely agree). The questionnaire has two subscales: physical activity subscale (FABQ-physical activity) and work subscale (FABQ-work), with four and seven items, respectively, and five remaining items as ineffective. Each subscale scores are summed to obtain a total score with possible scores of 0 to 24 for the FABQ-physical activity subscale and 0 to 42 for the FABQ- work subscale. Higher scores indicate greater fear-avoidance beliefs [45]. The Hausa version of the FABQ has been validated [46].

**Hausa Pain Catastrophizing Scale (PCS).** The PCS assesses thoughts and feelings about pain [47]. It consists of 13 items, with each item rated using a 5-point Likert scale ranging from 0 (not at all) to 4 (all the time). The scores obtained for each item are summed to obtain the total scores ranging from 0 to 52, with higher scores indicating more catastrophic thoughts [47]. The Hausa version of the PCS demonstrated acceptable reliability and construct validity [48].

**Hausa Oswestry Disability Index (ODI).** The ODI assesses perceived levels of functional disability [49]. It consists of ten categories, of which each is having six statements scored from 0 to 5. Scores obtained for each category are summed and divided by the number of completed categories to obtain a final score ranging from 0 to 100, with higher scores indicating greater disability [49]. The Hausa version of the ODI 2.1a has been validated [50].

**Hausa Short-form Health Survey (SF-12).** The SF-12 is a generic measure of health-related quality of life [51]. It consists of 12-item, designated into eight domains from which two global health constructs (physical and mental health) are derived. Each item is rated using response categories. The response categories vary from 2 to 6 and raw scores for items ranging from 1 to 6. A web-based scoring tool (www.orthotoolkit.com/sf-12/) was used to calculate the physical and mental component scores expressed in percentage, with higher scores indicating better health status. The Hausa version of the SF-12 has been validated [52].

**Hausa Visual Analogue Scale for pain (VAS-pain).** The VAS is widely used to assess levels of pain intensity. It consists of a 100mm horizontal line anchored on the left with the phrase "No Pain" and on the right with the phrase "Worst Imaginable Pain". A higher score indicates greater pain intensity [53]. Patients were asked to mark at a point on the line that corresponds to their current pain. The Hausa version of the VAS-pain was found to be reliable [54].

## Translation and cross-cultural adaptation

The translation and cross-cultural adaptation process followed the guidelines proposed by Beaton et al. [55] and the entire process consisted of six stages as follows:

1. Initial translation: Two independent bilingual (English and Hausa) translators, whose mother tongue is Hausa, forward translated the original English BBQ into Hausa, resulting in two versions (T1 and T2). The first translator (NBM) was a clinical physiotherapist and aware of the purpose of the study and the questionnaire concept whereas the second translator (TA) was a linguist and neither aware nor informed about the concept being examined.

2. Synthesis of the translation: The translators (T1 and T2) and the lead author discussed discrepancies of the translated versions using the original English version as reference. Following consensus, a synthesized version (T12) was then produced.

3. Back translation: Two independent bilingual translators (IMI and IU) with no medical or clinical background and blinded to the original English version translated the synthesized version (T12) back into English, resulting in two back translations (BT1 and BT2).

4. Expert committee review: An expert committee involving both the forward and backward translators, one academic physiotherapist (BK) with proficiency in methodology, and the lead author compared and consolidated all the translated versions (T1, T2, T12, BT1, BT2) taking into account achieving cross-cultural equivalence. This stage ensured face validity and resulted in the prefinal version of the Hausa BBQ for field-testing.

5. Test of the prefinal version: The prefinal version was tested among 20 patients (11 males and 9 females; mean age = 47 years) with non-specific CLBP and equal representation of urban (n = 10) and rural (n = 10) community. Each participant was asked to provide

feedback about clarity and interpretability of the questionnaire items and chosen responses. All problematic items, responses, statements, phrases, and words were resolved at this stage. The committee ensured that the original meaning of the questionnaire was not altered or lost while attaining cross-cultural equivalence. This stage ensured the content validity and led to the production of the final version of the Hausa BBQ (Hausa-BBQ).

6. Proofreading: A professional translator independently proofread the final version for any errors that may have been missed in the previous stages. The final version (see S1 Appendix) along with the reports of the translation process was then sent to the original developers of the questionnaire for appraisal. No further modifications were necessary.

### Assessment of psychometric properties

The procedure used throughout this section has been used in the cross-cultural adaptation of other Hausa self-report measures as reported elsewhere [48,50,52].

**Population.**   Generally, consensus about the ideal sample size for validating a scale is unclear. However, the Consensus-based Standards for the selection of health Measurement Instruments (COSMIN) checklist suggest that a sample size of $\geq 100$ patients is adequate for psychometric assessments [56]. In the present study, 200 patients were recruited to assess the psychometric properties of the Hausa-BBQ adequately.

The study was conducted purposely in one tertiary health facility (Murtala Muhammad Specialist Hospital [MMSH]), and three secondary health facilities (Dawakin-Kudu General Hospital [DGH], Wudil General Hospital [WGH], and Kura General Hospital [KGH]), all in Kano State, Northwestern Nigeria. These facilities were selected to recruit both urban and rural patients. The patients were consecutively recruited into the study at the physiotherapy outpatient unit in each of these facilities between February and May 2018. The inclusion criteria were adults aged between 18 to 70 years, with primary complain of non-specific LBP $\geq 12$ weeks and fluent in Hausa language. Participants were excluded if their LBP was due to serious spine pathologies such as fracture, infection, inflammatory disease, malignancy, and osteoporosis. Patients with a history of previous spine surgery, cognitive impairment or impaired capacity to be interviewed, and pregnancy were also excluded.

**Procedure for data collection.**   Four licensed physiotherapists with between two to five years of clinical experience were recruited from the selected health facilities and received a one-day training session on data collection procedures. The session was organized by the lead author for the physiotherapists to familiarize themselves with the collection of data using interviewer-administration method since a significant proportion of Hausa patients particularly rural dwellers are non-literates [50,52]. The physiotherapists, in each of the study settings, were responsible for eligibility assessments, which included history taking and screening of 'red flags to exclude serious spinal pathology. After applying informed consent, the participants' demographic and clinical characteristics were collected and recorded.

### Statistical analysis

To assess validity, 200 rural and urban patients completed the Hausa-BBQ along with the Hausa FABQ, PCS, ODI, SF-12 health survey, and VAS-pain. The measures were administered via interviewer-administration or self-administration method where applicable. To assess test-retest reliability, the Hausa-BBQ was re-administered among 100 patients who participated in the validity testing. Measurements were repeated 7 to 14 days after the initial measurement.

All statistical analyses were conducted using IBM SPSS for Windows version 24.0 (IBM Corp, Armonk, NY). Normality of the data was assessed using visual inspection of distribution

plots, and Kolmogorov-Smirnov and Shapiro-Wilk's tests. Descriptive statistics of mean, standard deviation (SD), frequencies, and percentages were used to summarize the data. Specific statistical techniques used to assess the psychometric properties of the Hausa-BBQ were as follows:

**General aspects and ceiling and floor effects.** Potential missing values were evaluated by cross-checking all the items to ensure that respondents did not leave any item unanswered. Ceiling or floor effects are considered if more than 15% of respondents scored the maximum or minimum possible scores, respectively [57]. Potential "ceiling and floor effects were assessed by calculating the percentage of the patients obtaining the maximum (ceiling) or minimum (floor) BBQ scores.

**Internal construct validity.** Factorial structure was assessed using exploratory factor analysis (EFA) by applying principal component analysis with orthogonal Varimax rotation. Kaiser-Meyer-Olkin (KMO) test and Bartlett's test of sphericity were applied to determine sampling adequacy for appropriateness of the factor analysis. A significant Bartlett's test ($p < 0.05$) and KMO value of $> 0.6$ were considered acceptable [58]. Since the BBQ is well known to be a unidimensional scale, we hypothesized that the factorial validity of the Hausa-BBQ would be supported if the nine scoring items (9-item scores) loaded significantly (factor loading coefficients $\geq 0.4$) on one underlying factor [36,38].

**External construct validity.** Convergent and divergent validity were assessed by correlating the Hausa-BBQ scores with the scores of other measures. Predefined hypotheses of association were formulated based on the findings of previous validation studies. For the convergent validity, we hypothesized that the BBQ would have low to strong negative correlations with the FABQ-physical activity ($r$ or $rho = -0.30$ to $-0.57$) [34,35,38] and FABQ-work ($r$ or $rho = -0.29$ to $-0.55$) [34,38]; low to moderate negative correlations with the PCS ($r$ or $rho = -0.30$ to $-0.50$) [35] and ODI ($r$ or $rho = -0.30$ to $-0.42$) [33,36,38]; and low to moderate positive correlations with the PCS-12 ($r$ or $rho = 0.28$ to $0.50$) [33] and MCS-12 ($r$ or $rho = 0.23$ to $0.50$) [33]. For the divergent validity, we hypothesized that the BBQ would correlate weakly with the VAS-pain ($rho = -0.14$ to $-0.34$) [33,35,38]. The construct validity is supported when at least 75% ($\geq 5$) of the predefined hypotheses are confirmed [57]. Pearson's correlation coefficient ($r$) was applied for normally distributed variables while Spearman's correlation coefficient ($rho$) was applied for non-normally distributed variables. Correlation coefficients were interpreted as "strong/high" (0.51–1.00), "moderate" (0.31–0.50), and "weak/low" (0.10–0.30) [59,60].

Known-groups validity was assessed by comparing the mean Hausa-BBQ scores of different patient groups based on age (younger adults: 18–24; adults: 25–44; midlife adults: 45–64, and older adults: $\geq 65$ years) [52] and education (non-formal education, completed primary education, completed secondary education, and completed tertiary education) [48,50,52] using one-way analysis of variance (ANOVA). We hypothesized that younger [25] and non-literate (non-formal education) [24] patients would have more pessimistic beliefs about back pain.

**Item analysis.** The item analysis included inter-item correlations, corrected item-total correlations, and Cronbach's α if item deleted. Inter-item correlations and corrected item-total correlations were examined to determine item redundancy, with correlations of 0.30–0.70 being considered satisfactory [61]. Pearson's correlation coefficient was applied for the inter-item correlations Cronbach's α if item deleted was calculated to determine internal consistency (homogeneity of items). Cronbach's α coefficients were interpreted as "inadequate" ($< 0.70$), "adequate" (0.70–0.79), "good" (0.80–0.89), and "excellent" ($> 0.90$) [62]. We hypothesized that the Cronbach's α coefficients for the Hausa-BBQ would lie within the range of 0.70–0.82 to be considered acceptable [31,33,36,40,42].

**Test-retest reliability.** Test-retest reliability (temporal stability) was assessed by calculating ICC with 95% confidence intervals (CI) for agreement. ICC values were interpreted as

"poor" ($< 0.40$), "fair" (0.40–0.59), "good" (0.60–0.74), and "excellent" ($> 0.75$) [62]. We hypothesized that the ICC for the Hausa-BBQ would lie within the range of 0.70–0.89 to be considered acceptable [31,32,35,37,41].

As per the recommendation of COSMIN [63], absolute reliability was assessed by calculating standard error of measurement (SEM) and minimal detectable change at 95% CI (MDC$_{95}$). The SEM was calculated by taking the square root of the mean square error term from the reliability ANOVA output. Subsequently, the MDC$_{95}$ was calculated by multiplying the SEM by 2.77. The MDC$_{95}$ provides the minimum values considered true change beyond measurement error [64,65]. We hypothesized that the SEM and MDC$_{95}$ for the Hausa-BBQ would lie within the range of 2.1–3.8 [33,36,42] and 5.9–10.5 [35,36,42], respectively. Additionally, 95% limits of agreement (LOA$_{95\%}$) were calculated with Bland-Altman plots by plotting the difference between test and retest of Hausa-BBQ scores against the mean scores of the test and retest. This was done for both the global and 9-item scores. We hypothesized that the LOA$_{95\%}$ for the Hausa-BBQ would lie within the range of –10.0 to +12.6 [31,34,42]. Subgroup reliability analyses regarding ICC, SEM and MDC$_{95}$ for urban and rural patients were also performed.

## Results

### Translation and cross-cultural adaptation

The Hausa-BBQ was easily adapted as there were no major forward or backward translation issues. However, during the pilot testing, some patients expressed difficulty in choosing the applicable response number as the original English version has only two descriptors; completely disagree (1) and completely agree (5). Hence, for clarity, the expert committee reached a consensus to add descriptors "disagree", "neutral", and "agree" for response numbers 2, 3, and 4, respectively. The expert committee ensured that the Hausa-BBQ attained semantic, idiomatic, experiential, and conceptual equivalence with the original English version. All the questionnaire items were reported to be clear and comprehensive.

### Psychometric assessment

**Demographic and clinical characteristics.** The mean age of the patients was 45.5±14.5 years. There were 123 (61.5%) males and 77 (38.5%) females. The majority of the patients were residing in rural areas (60.0%) and unemployed (64.0%). Slightly over half of them had non-formal education (33.0%) and were non-literates in Hausa (55.5%). The demographic and clinical characteristics of the patients across validity and reliability are fully presented in Table 1.

**General aspects and ceiling and floor effects.** All the participants completed the Hausa-BBQ without missing values yielding a response rate of 100%. The response rate of the questionnaire according to the recruitment sites were 78 (39.0%), 47 (23.5%), 39 (19.5%), and 36 (18.0%) in the MMSH, DGH, WGH, and KGH, respectively. No ceiling and floor effects were observed for the questionnaire global scores or 9-item scores. However, for the individual items, ceiling effects (higher scores) were found for item 4, whereas floor effects (lower scores) were found for items 2, 5, 7, 9, 11, 12, 13, and 14 (Table 2). The response trend for each item of the Hausa-BBQ does not deviate from normal distribution as none of the items exhibited skewness $> 1.96$ (Table 2).

**Internal construct validity.** The KMO value was adequate (0.754) and Bartlett's test of sphericity was significant ($\chi^2 = 794.8$, df = 91, $p = 0.000$) signifying appropriateness of the factor analysis. The principal component analysis revealed a 4-factor structure explaining 58.9% of the total variance (Table 3). The nine scoring items loaded on factor 1 except for item 8, which loaded on factor 3. The distractor items loaded on factor 2 except item 4, which loaded on factor 4. None of the distractor items loaded on the same factors as the nine scoring items

Table 1. Demographic and clinical characteristics of the study population.

| Characteristics | Validity (n = 200) | Reliability (n = 100) |
|---|---|---|
| Age, years, mean ± SD | 45.5 ± 14.5 | 46.3 ± 14.7 |
| Gender, *n (%)*, male: female | 123 (61.5), 77 (38.5) | 61 (61.0), 39 (39.0) |
| Habitation, *n (%)*, urban: rural | 80 (40.0), 120 (60.0) | 42 (42.0), 58 (58.0) |
| Marital status, *n (%)*, married: unmarried | 157 (78.5), 43 (21.5) | 80 (80.0), 20 (20.0) |
| Educational status, *n (%)* | | |
| Non-formal education | 66 (33.0) | 32 (32.0) |
| Completed primary education | 30 (15.0) | 16 (16.0) |
| Completed secondary education | 41 (20.0) | 18 (18.0) |
| Completed tertiary education | 63 (31.5) | 34 (34.0) |
| Literacy (ability to read and write in Hausa), *n (%)* | | |
| Non-literate | 111 (55.5) | 54 (54.0) |
| Literate | 89 (44.5) | 46 (46.0) |
| Occupational status, *n (%)* | | |
| Employed | 49 (24.5) | 23 (23.0) |
| Unemployed | 128 (64.0) | 65 (65.0) |
| Student | 17 (8.5) | 10 (10.0) |
| Retiree | 6 (3.0) | 2 (2.0) |
| BBQ, mean ± SD (global score, range 14–70) | 36.0±7.24 | 37.3±5.70 |
| BBQ, mean ± SD (9-item score, range 9–45) | 23.2±5.42 | 23.9±5.23 |
| FABQ-physical activity, mean ± SD (score range 0–42) | 13.1±5.80 | - |
| FABQ-work, mean ± SD (score range 0–24) | 23.4±7.77 | - |
| PCS, mean ± SD (score range 0–52) | 30.0±8.21 | - |
| PCS-12, mean ± SD (score range 0–100) | 34.5±6.94 | - |
| ODI, mean ± SD (score range 0–100) | 37.2±13.2 | - |
| MCS-12, mean ± SD (score range 0–100) | 38.8±10.1 | - |
| VAS-pain, mean ± SD (score range 0–100mm) | 41.3±13.1 | - |

SD, standard deviation; BBQ, Back Beliefs Questionnaire; FABQ, Fear-avoidance Beliefs Questionnaire; PCS, Pain Catastrophizing Scale; ODI Oswestry Disability Index; PCS-12, Physical Component Summary; MCS-12, Mental Component Summary; VAS-pain, Visual Analogue Scale for pain.

(Table 3). The internal consistency as measured by the Cronbach's α of the nine scoring items was 0.72, and 0.79 after the removal of item 8 indicating that the items still maintain homogeneity within the scale.

**External construct validity.** The normality analyses revealed that the BBQ, PCS-12, and MCS-12 were normally distributed whereas the FABQ-PA, FABQ-W, PCS, and VAS-pain had skewed distribution. Consistently with the convergent hypotheses, the Hausa-BBQ demonstrated a low negative correlation with FABQ-PA ($rho = -0.18$, $p > 0.012$), FABQ-W ($rho = -0.21$, $p > 0.003$), and PCS ($rho = -0.20$, $p > 0.004$) except with the ODI ($rho = -0.21$, $p > 0.003$); and a low positive correlation with the PCS-12 ($r = 0.24$, $p > 0.001$) and MCS-12 ($r = 0.23$, $p > 0.001$). As for the divergent hypotheses, the Hausa-BBQ demonstrated a low negative correlation with the VAS-pain ($rho = -0.19$, $p > 0.006$) as expected. The correlational analyses indicate that 85% (6:7) of the predefined hypotheses were confirmed.

Known-groups comparison showed that the Hausa-BBQ did not significantly discriminate between patients with different age groups ($p > 0.05$) (Table 5). However, the questionnaire significantly discriminates between patients with different education levels ($p < 0.05$). Patients with non-formal education demonstrated lower BBQ scores (Table 4).

**Table 2. General characteristics of the Hausa Back Beliefs Questionnaire (n = 200).**

|    |                                                                | Range | Mean (SD)   | Ceiling effects n (%) | Floor effects n (%) | Skewness |
|----|----------------------------------------------------------------|-------|-------------|-----------------------|---------------------|----------|
| 1  | There is no real treatment for back trouble                    | 1–5   | 2.66 (0.88) | 10 (5.0)              | 16 (8.0)            | 0.457    |
| 2  | Back trouble will eventually stop you from working             | 1–5   | 2.77 (1.33) | 30 (15.0)             | 47 (23.5)           | 0.218    |
| 3  | Back trouble means periods of pain for the rest of one's life  | 1–5   | 2.69 (0.98) | 11 (5.5)              | 26 (13.0)           | 0.190    |
| 4  | Doctors cannot do anything for back trouble                    | 1–5   | 3.06 (1.36) | 48 (24.0)             | 28 (14.0)           | 0.131    |
| 5  | A bad back should be exercised                                 | 1–5   | 2.51 (1.47) | 18 (9.0)              | 85 (42.5)           | 0.245    |
| 6  | Back trouble makes everything in life worse                    | 1–5   | 2.52 (0.82) | 2 (1.0)               | 26 (13.0)           | −0.214   |
| 7  | Surgery is the most effective way to treat back trouble        | 1–5   | 2.42 (1.13) | 11 (5.5)              | 57 (28.5)           | 0.325    |
| 8  | Back trouble may mean you end up in a wheelchair.              | 1–5   | 2.72 (0.96) | 6 (3.0)               | 29 (14.5)           | −0.117   |
| 9  | Alternative treatments are the answer to back trouble          | 1–5   | 2.30 (1.16) | 7 (3.5)               | 69 (34.5)           | 0.419    |
| 10 | Back trouble means long periods of time off work               | 1–5   | 2.68 (0.95) | 6 (3.0)               | 24 (12.0)           | 0.055    |
| 11 | Medication is the *only* way of relieving back trouble         | 1–5   | 2.53 (1.17) | 18 (9.0)              | 71 (35.5)           | 0.275    |
| 12 | Once you have had back trouble there is always a weakness       | 1–5   | 2.41 (0.94) | 2 (1.0)               | 42 (21.0)           | −0.029   |
| 13 | Back trouble *must* be rested                                  | 1–5   | 2.28 (0.83) | 1 (0.5)               | 39 (19.5)           | 0.012    |
| 14 | Later in life back trouble gets progressively worse            | 1–5   | 2.54 (0.98) | 9 (4.5)               | 33 (16.5)           | 0.282    |
|    | Global scores                                                  | 14–70 | 36.0 (7.24) | 60 (0.5)              | 18 (0.5)            | −0.062   |
|    | 9-item scores                                                  | 9–45  | 23.2 (5.42) | 39 (0.5)              | 10 (0.5)            | −0.033   |

SD, standard deviation.

**Item analysis.** The inter-item correlations were < 0.70, except for the correlation between item 3 and 14 ($r = 0.73$), suggesting multicollinearity. However, since the correlation between these items was not considerably high and deleting any of the items significantly reduced the Cronbach's α, we decided to retain all the items to maintain the scale structure. The scale's corrected item-total correlations were 0.14–0.57, with low corrected item-total correlations being observed for item 8 (< 0.30), indicating redundancy. The Cronbach's α if item deleted was 0.75. Deletion of item 8 slightly increased the Cronbach's α (0.78) (Table 5).

**Test re-test reliability.** As shown in Table 5, the ICC for the overall population was excellent (ICC = 0.91; 95% CI = 0.86–0.94), with minimal SEM and $MDC_{95}$ (1.9 and 5.2, respectively).

**Table 3. Factor structure of the Hausa Back Beliefs Questionnaire.**

| Statement |                                                                | Coefficients ≥ 0.4 | | | |
|-----------|----------------------------------------------------------------|----------|----------|----------|----------|
|           |                                                                | Factor 1 | Factor 2 | Factor 3 | Factor 4 |
| 1         | There is no real treatment for back trouble                    | **0.486***  | 0.048    | 0.614    | 0.104    |
| 2         | Back trouble will eventually stop you from working             | **0.600***  | 0.140    | −0.283   | 0.120    |
| 3         | Back trouble means periods of pain for the rest of one's life  | **0.797***  | −0.021   | 0.219    | −0.108   |
| 4         | Doctors cannot do anything for back trouble                    | 0.051    | 0.193    | −0.147   | **0.750*** |
| 5         | A bad back should be exercised                                 | 0.034    | **0.432*** | −0.258   | −0.648   |
| 6         | Back trouble makes everything in life worse                    | **0.519***  | 0.143    | 0.338    | 0.163    |
| 7         | Surgery is the most effective way to treat back trouble        | 0.125    | **0.730*** | 0.117    | 0.402    |
| 8         | Back trouble may mean you end up in a wheelchair.              | 0.129    | 0.081    | **0.781*** | −0.064   |
| 9         | Alternative treatments are the answer to back trouble          | 0.124    | **0.834*** | 0.050    | 0.023    |
| 10        | Back trouble means long periods of time off work               | **0.575***  | 0.061    | 0.252    | −0.053   |
| 11        | Medication is the *only* way of relieving back trouble         | −0.025   | **0.848*** | 0.039    | −0.156   |
| 12        | Once you have had back trouble there is always a weakness       | **0.537***  | 0.117    | 0.187    | 0.050    |
| 13        | Back trouble *must* be rested                                  | **0.696***  | 0.018    | −0.101   | 0.140    |
| 14        | Later in life back trouble gets progressively worse            | **0.757***  | −0.065   | 0.257    | −0.144   |
|           | % variance explained                                           | 27.1     | 15.1     | 9.0      | 7.7      |

**Table 4. Known-groups validity of the Hausa Back Beliefs Questionnaire.**

| | Age group | | | | | | |
|---|---|---|---|---|---|---|---|
| | 18–24 | 25–44 | 45–64 | ≥ 65 | | | |
| | Mean (SD) | Mean (SD) | Mean (SD) | Mean (SD) | F-ratio | *p*-value | ηp2 |
| BBQ (9–45) | 24.6 (5.40) | 23.4 (5.22) | 22.9 (5.31) | 22.9 (6.38) | 0.532 | 0.661 | 0.01 |
| | Education level | | | | | | |
| | Non-formal | Primary | Secondary | Tertiary | | | |
| | n (%) | n (%) | n (%) | n (%) | F-ratio | *p*-value | ηp2 |
| BBQ (9–45) | 21.6 (5.63) | 24.0 (6.24) | 23.8 (5.02) | 24.1 (4.69) | 2.951 | 0.034* | 0.04 |

BBQ, Back Beliefs Questionnaire; SD, standard deviation; ηp2, partial eta squared.

*$p < 0.05$.

The mean difference (–0.30) of the repeated measurements was not statistically significant ($p > 0.05$). As for the subgroup analyses, the ICC, SEM, and $MDC_{95}$ calculated for the urban (n = 42) and rural (n = 58) populations were comparable to the overall population (Table 6). The Bland-Altman analysis for the overall population showed a mean difference and $LOA_{95\%}$ of –0.30 and –5.11 to +5.71 (Fig 1).

## Discussion

The original English BBQ was successfully adapted into Hausa without major translation problems similar to many previous adaptations [36–38]. The questionnaire was comprehensive, clear, and easy to complete thus demonstrating good face and content validity. The absence of missing values could be ascribed to the interviewer-administration method used in the study, besides the effort of the raters to ensure that none of the questionnaire items was left unanswered including those who completed the questionnaire in self-administered format. No ceiling neither floor effects were detected for both the global and 9-item scores. However, ceiling effects were found for item 4 as most patients disagree that doctors cannot do anything for back trouble. Likewise, floor effects were detected for items 2, 5, 7, 9, 11, 12, 13, and 14 indicating poor discriminating ability as most patients have lower scores (more pessimistic beliefs about LBP) in these items. These findings could be partly explained by the fact that a plurality of the studied population (33.0%) had non-formal education. In line with our study, ceiling and floor effects were found in some of the items of the French adaptation [39].

Regarding factorial validity, only a few studies [36,38,41,42] have examined the underlying structure of the BBQ (Table 6) even though the questionnaire was originally designed as a unidimensional scale [20]. Though our factor analysis revealed a four-factor solution, surprisingly, the one-factor structure is supported as the nine scoring items loaded on the first factor except

**Table 5. Internal consistency and test-retest reliability of the Hausa Back Beliefs Questionnaire.**

| Hausa-BBQ | Internal consistency | Test-retest (repeatability) | | | | SEM | MDC95 |
|---|---|---|---|---|---|---|---|
| | Cronbach's α if item deleted | Test (t1) Mean SD | Retest (t2) Mean SD | t1-t2 | ICC (95% CI) | | |
| Overall population | 0.78 | 23.9 (5.23) | 24.2 (4.46) | –0.30† | 0.91 (0.86–0.94) | 1.9 | 5.2 |
| Urban population | - | 23.4 (3.92) | 24.1 (3.97) | –0.71† | 0.89 (0.80–0.94) | 1.7 | 4.6 |
| Rural population | - | 24.2 (6.01) | 24.2 (4.81) | –0.00† | 0.91(0.86–0.95) | 2.1 | 5.9 |

BBQ, Back Beliefs Questionnaire; ICC, intraclass correlation coefficient; CI, confidence interval; SEM, standard error of measurement; MDC, minimal detectable change.

†$p > 0.05$.

**Table 6. Summary of psychometric properties of the published adapted Back Beliefs Questionnaire.**

| First author, year | Adapted to | n[a] | Int. cons. | Test-retest reliability | | | | | n[b] | Construct validity (r or rho)[c] | Factor Analysis | |
|---|---|---|---|---|---|---|---|---|---|---|---|---|
| | | | α | Days | ICC | SEM | MDC | LOA | | Measure | Model | R² |
| Teixeira, 2020 [31] | Brazilian Portuguese | 26 | 0.70 | 7–14 | 0.74 | 4.0 | 11.1 | −10.5 to +12.0 | 26 | - | - | - |
| Mbada, 2020 [42] | Yoruba (Nigerian) | 51 | 0.71 | 7 | 0.89 | 2.3 | 6.4 | −0.684 to +5.70 | 119 | VAS = 0.27 | 3, 2 | 44.9, 36.2 |
| Rajan, 2020 [37] | Marathi (Indian) | 43 | 0.67 | 15 | 0.80 | - | - | | 50 | RMDQ = −0.29 | - | - |
| Tingulstad, 2019 [35] | Norwegian | 63 | 0.82 | 1–13 | 0.71 | 3.8 | 10.5 | | 116 | NRS = −0.14; RMDQ = −0.29; FABQ-PA = −0.57; PCS = −0.45 | - | - |
| Karaman, 2019 [38] | Turkish | 25 | 0.79 | 7 | 0.84 | - | - | | 110 | NRS = −0.34; ODI = −0.42; FABQ-PA and W = −0.55; HADS-anxiety = −0.46; HADS-depression = −0.32 | 3 | 52 |
| Cheung, 2018 [33] | Traditional Chinese (Hong Kong) | 100 | 0.81 | - | - | - | - | | 100 | VAS = − 0.32; ODI = −0.34; FABQ-PA = −0.34; FABQ-W = − 0.29; PF = 0.27; RP = 0.39; BP = 0.22; GH = 0.32; VT = 0.30; SF = 0.28; RE = 0.27; MH = 0.24; PCS-12 = 0.28; MCS-12 = 0.23 | - | - |
| Dupeyron, 2017 [39] | French | 121 | 0.80 | 1–7 | 0.64 | - | - | | 128 | VAS = −0.15; Tampa = −0.66; FABQ = −0.52; Quebec = −0.31; Dallas = −0.24 to −0.43; HADS-anxiety = −0.28; HADS-depression = −0.42 | - | - |
| Maki, 2017 [34] | Modern Arabic (Bahrain) | 64 | 0.73 | 7 | 0.80 | - | - | −8.00 to +12.6 | 199 | FABQ = −0.33; FABQ-PA = −0.30; FABQ-W = −0.29 | - | - |
| Alamrani, 2016 [36] | Arabic (Saudi Arabia) | 25 | 0.77 | 1–8 | 0.88 | 2.1 | 5.9 | | 115 | NRS = −0. 10; ODI = −0.31 | 3 | 46 |
| Elfering, 2015 [41] | German | 151 | 0.80 | 4–13 | 0.89 | - | - | | 2225 | - | 3 | 48 |
| Suzuki, 2012 [40] | Japanese | - | 0.82 | - | - | - | - | | 127 | WPAI = −0.26; RMDQ = −0.20; NRS = −0.04 | - | - |
| Chen, 2011 [32] | Simplified Chinese (Shanghai) | 65 | 0.70 | 1–10 | 0.85 | - | - | | 65 | VAS = −0.04; HC-PAIRS = 0.40; FABQ-PA = 0.48; FABQ-W = 0.49 | - | - |

Int. cons, internal consistency; α, Cronbach's alpha; ICC, intraclass correlation coefficient; SEM, standard error of measurement; MDC, minimal detectable change; VAS, Visual Analogue Scale; RMDQ, Roland-Morris Disability Questionnaire; NRS, Numerical Rating Scale; ODI, Oswestry Disability Index; FABQ-PA, Fear-Avoidance Beliefs Questionnaire-Physical activity; FABQ-W, Fear-Avoidance Beliefs Questionnaire-work; PCS, Pain Catastrophizing Scale; HADS, Hospital Anxiety and Depression Scale, PF, Physical Functioning; RP, Role Physical; BP, Bodily Pain; GH, General Health; VT, Vitality; SF, Social Functioning; RE, Role Emotional; MH, Mental Health; PCS-12, Physical Component Summary; MCS-12, Mental Component Summary; HC-PAIRS, Health Care Providers' Pain and Impairment Relationship Scale; WPAI, Work Productivity and Activity Impairment Questionnaire.

[a]Test-retest reliability sample size.

[b]Construct validity and internal consistency (α) sample size.

[c]Measures used to evaluate construct validity of BBQ using Pearson's product correlation (r) or Spearman's rank correlation (rho).

R²Total variance explained.

for item 8 "Back trouble may mean you end up in a wheelchair", which loaded on the third factor. The result that item 8 was not loaded on the first factor might be attributed to the different interpretations for the item among the studied population. As expected, the distractor items loaded on the second factor except item 4 "Doctors cannot do anything for back trouble", which loaded on the fourth factor. The finding that the one-factor structure is supported is in line with the original English version [20]. Although studies exploring the factor structure of the BBQ consistently revealed a three-factor structure [30,36,38,41,42], interestingly, most of these studies [36,38,41] found the nine scoring items loaded on the first factor solution while

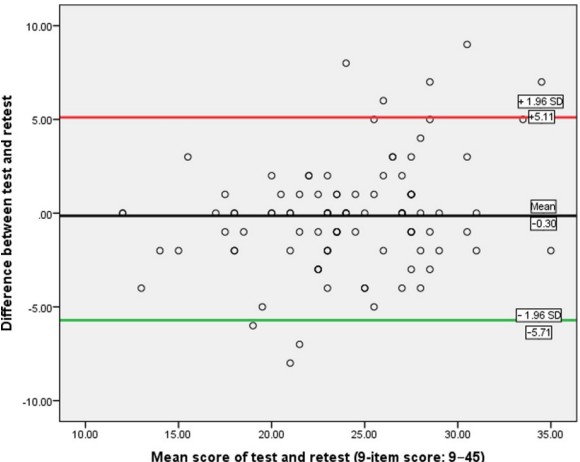

**Fig 1. Bland-Altman plot for test-retest agreement of Hausa-BBQ.**

the distractors loaded on the two other factor solution resembling the findings of the present study.

To examine convergent and divergent validity of the Hausa-BBQ, various specific hypotheses were constructed, and on this basis, the construct validity is supported as 85% (6 out of 7) of the predefined hypotheses were confirmed. Overall, the convergent validity of the Hausa-BBQ was demonstrated, with significant correlations with the FABQ-PA, FABQ-W, PCS, PCS-12, and MCS-12. While the BBQ and FABQ scales measure beliefs about LBP, the low correlations observed between these scales may be attributed to the fact that they measure a construct that is not similar to each other [36]. The correlation between the Hausa-BBQ and ODI (*rho* = −0.21) was not at least moderate as hypothesized, even though was significant. It is anticipated that low knowledge about LBP would be associated with more disability. The low correlations observed can be explained by the inverse relationship between these scales, besides only 25% of the studied population were workers. Hence, it can be speculated that unemployed patients may not be concerned with back pain consequences beliefs related to work. To the best of our knowledge, only the Norwegian adaptation [35] assessed the correlation between the BBQ and PCS (Table 6). In the present study, the correlation (*rho* = −0.24) between these measures was lower than that found for the Norwegian version (*rho* = −0.45) [35]. Similar to the Traditional Chinese adaptation [33], low positive correlations were found between the Hausa-BBQ and the PCS-12 and MCS-12, hence, establishing evidence of the association between pessimistic beliefs about LBP and physical as well as mental health. Further, the low negative correlation observed with the VAS-pain confirms discriminative validity. Thus, it can be inferred that beliefs about negative consequences of LBP may not be exclusively related to pain intensity. In line with our study, low or no correlations (*r* or *rho* = −0.04 to −0.15) between back beliefs and pain intensity were generally reported in the literature [32,35,36,39,40,66].

The results of the known-groups validity revealed that patients with non-formal education had lower BBQ scores, which implies more pessimistic beliefs about LBP, and the ability of the questionnaire to discriminate well for patients who differed in education. This is in concordance with previous studies demonstrating an association between back pain beliefs and education levels [24,25]. Clinicians should therefore consider reshaping patients' beliefs about LBP using effective education strategies. On the contrary, we found no significant relationship between age groups and the BBQ scores suggesting that the questionnaire was unable to

discriminate against patients who differed in age. In line with this finding, previous studies did not find age to be an important correlate of back beliefs [16,24].

Internal consistency calculated for the Hausa-BBQ ($\alpha$ = 0.78) was adequate considering the acceptable value of 0.70 [57]. Our alpha coefficient is slightly higher than the 0.70 reported for the original English measure [20], and the range of 0.67–0.77 reported by many language versions [31,32,34,36,37,42,43]. Other adaptations, however, reported slightly higher alpha coefficients ($\alpha$ range = 0.79–0.82) [33,35,38,39,40] (Table 6). Though item 8 shows redundancy (weak correlation < 0.30) as also revealed by the factor analysis, deletion of this item, however, did not significantly improve the internal consistency ($\alpha$ = 0.75 *vs.* 0.78), hence, this item may still be included in computing the scores of the Hausa-BBQ to retain the original structure of the questionnaire. Moreover, the multicollinearity (high correlation > 0.70) detected between items 3 and 14 suggests that these items may be measuring the same aspect of inevitable beliefs.

Test-retest reliability of the Hausa-BBQ demonstrated a highly significant correlation (ICC = 0.91), suggesting acceptable reliability. Remarkably, our ICC was higher than the value (0.87) obtained for the original English version [20] and the range of 0.64–0.89 reported by several translated versions [31,32,34–38,41,42] (Table 6). Additionally, the ICC values obtained for the urban and rural subgroups in the present study were also excellent and comparable to those obtained for the overall population. This suggests that our questionnaire is reliable when used in different contexts. It should be noted, however, that ICC only takes into account between-subject variability but not measurement error [67]. Only four of the twelve BBQ adaptations (Table 6) calculated measurement error expressed as SEM and the resultant $MDC_{95}$. In the present study, these reliability indicators were minimal (SEM = 1.9; $MDC_{95}$ = 5.2) when compared to the Brazilian-Portuguese (SEM = 4.0; $MDC_{95}$ = 11.1) [31] and Norwegian (SEM = 3.8; $MDC_{95}$ = 10.5) [35] versions but somewhat comparable to the Arabic (SEM = 2.1; $MDC_{95}$ = 5.9) [36] and Yoruba (SEM = 2.3; $MDC_{95}$ = 6.4) [42] versions. Also, the SEM and $MDC_{95}$ values obtained for the urban and rural subgroups were comparable to those obtained for the overall population, thus supporting the applicability of the Hausa-BBQ in both rural and urban Nigeria. The $MDC_{95}$ is an essential measurement property as it indicates a true change in a patient's score beyond measurement error. For example, when using the Hausa-BBQ (0–45) as an outcome measure, an observed change greater than 5.2 points can be considered a real change whereas an observed change less than 5.2 points cannot be distinguished from measurement error.

Although the Bland-Altman plot method does not reveal whether the limits are acceptable but defines the intervals of agreements, the smaller the range between two limits the better the agreement. The $LOA_{95\%}$ calculated for the Hausa-BBQ showed a good distribution of scores as the mean difference was close to zero with few outliers. Compared to the $LOA_{95\%}$ range of –10.0 to +12.6 observed in previous validations [31,34,42], our range (−5.71 to +5.11) was smaller, suggesting good agreement with minimal systematic bias. This indicates that researchers and clinicians can have confidence when administering the Hausa-BBQ that the measurements will not be diluted by systematic bias or random error. Moreover, given that the minimal important change (MIC) for the BBQ has not been determined, the $MDC_{95}$ and the $LOA_{95\%}$, though should not replace the MIC, can be used to interpret a change in BBQ scores following interventions targeting negative beliefs about LBP.

The strength of this study is that the psychometric assessment of the Hausa-BBQ is in line with COSMIN guidelines [56,63]. Moreover, both rural and urban patients with different literacy levels were recruited to have wide applicability of the questionnaire. However, one potential limitation of this study is that we could not guarantee that some patients did not receive or seek treatment during the recruitment or retesting period, which may influence their back beliefs. The majority of the respondents were interviewer-administered which might increase

measurement error or overestimate the clearness and readability of the questionnaire. Further-more, we did not perform confirmatory factor analysis, and responsiveness (sensitivity to change) to determine MIC. Future studies are therefore needed to evaluate these important psychometric properties.

## Conclusions

The results of this study suggest that the Hausa-BBQ was successfully adapted and psychometrically sound in terms of internal and external construct validity, internal consistency, and test-retest reliability in mixed urban and rural Hausa-speaking populations with chronic LBP. The questionnaire can be used to detect and categorize specific attitudes and beliefs about back pain in Hausa culture to prevent or reduce potential disability due to LBP.

## Supporting information

**S1 Appendix. Hausa version of BBQ.**
(PDF)

**S1 Data. The Hausa-BBQ validity data (n = 200).**
(XLSX)

**S2 Data. The Hausa-BBQ reliability data (n = 100).**
(XLSX)

## Acknowledgments

The authors would like to thank all the translators for their support during the translation and cross-cultural adaptation process, the patients who participated in this study, and the personnel especially Dr. Bashir Bello Abdullahi (PT) who assisted in the validation process.

## Author Contributions

**Conceptualization:** Aminu Alhassan Ibrahim, Mukadas Oyeniran Akindele.

**Data curation:** Aminu Alhassan Ibrahim, Bashir Kaka.

**Formal analysis:** Aminu Alhassan Ibrahim, Bashir Kaka, Bashir Bello.

**Investigation:** Aminu Alhassan Ibrahim, Mukadas Oyeniran Akindele.

**Methodology:** Aminu Alhassan Ibrahim, Mukadas Oyeniran Akindele, Bashir Kaka, Bashir Bello.

**Project administration:** Mukadas Oyeniran Akindele, Sokunbi Oluwaleke Ganiyu.

**Resources:** Aminu Alhassan Ibrahim.

**Software:** Bashir Bello.

**Supervision:** Mukadas Oyeniran Akindele, Sokunbi Oluwaleke Ganiyu.

**Validation:** Aminu Alhassan Ibrahim, Bashir Kaka, Bashir Bello.

**Visualization:** Mukadas Oyeniran Akindele, Sokunbi Oluwaleke Ganiyu, Bashir Kaka.

**Writing – original draft:** Aminu Alhassan Ibrahim.

**Writing – review & editing:** Aminu Alhassan Ibrahim, Sokunbi Oluwaleke Ganiyu, Bashir Kaka, Bashir Bello.

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
