## [Decision Letter · Decision Letter 0]

5 Jan 2021

PONE-D-20-31974

The Hausa back beliefs questionnaire: Translation, cross-cultural adaptation and psychometric assessment in mixed urban and rural Nigerian populations with chronic low back pain

PLOS ONE

Dear Dr. Ibrahim,

Thank you for submitting your manuscript to PLOS ONE. After careful consideration, we feel that it has merit but does not fully meet PLOS ONE’s publication criteria as it currently stands. Therefore, we invite you to submit a revised version of the manuscript that addresses the points raised during the review process.

The reviewers were generally positive and found merit in the science of the research but they still raised some important questions that need to be satisfactorily addressed in your revision before editorial decision on publication could be made. I encourage you to make all recommended changes as much as possible.

We look forward to receiving your revised manuscript.

Kind regards,

Adewale L. Oyeyemi, Ph.D

Academic Editor

PLOS ONE

Journal Requirements:

2. Please include an English translation of the Hausa questionnaire, as Supporting Information, or include a citation if it has been published previously.

3.Thank you for stating the following financial disclosure:

 "NO. The funders had no role in study design, data collection and analysis, decision to publish, or preparation of the manuscript."

Reviewers' comments:

Reviewer's Responses to Questions

**Comments to the Author**

1. Is the manuscript technically sound, and do the data support the conclusions?

Reviewer #1: Yes

Reviewer #2: Partly

Reviewer #3: Yes

2. Has the statistical analysis been performed appropriately and rigorously? 

Reviewer #1: Yes

Reviewer #2: Yes

Reviewer #3: Yes

3. Have the authors made all data underlying the findings in their manuscript fully available?

Reviewer #1: Yes

Reviewer #2: Yes

Reviewer #3: Yes

4. Is the manuscript presented in an intelligible fashion and written in standard English?

Reviewer #1: Yes

Reviewer #2: Yes

Reviewer #3: Yes

5. Review Comments to the Author

Reviewer #1: I reviewed this validation study of a nice scale like BBQ. There are some parts that need to be corrected. You can find my suggestions below.

1)General

Please review for spelling and grammatical mistakes.

2) Abstract

It's been too long. it should be summed up more briefly.

3)İntroduction

The paragraph that starts as “Most developed patient-reported… “ and gives general information about PROMs contains unnecessary information. This paragraph can be omitted.

“Nigeria is Africa's most populous….” This paragraph as was also lengthened unnecessarily. In this paragraph, the necessity of the hausa version should be stated more briefly.

4) Methods

Cronbach and ICC values given for the original BBQ are not in excelent ranges. Look again, which value corresponds to which range.

No reference is given to the PCS hausa version.

The first paragraph in “Assessment of outcomes” was also very unnecessary. We can take it out in this part.

The ranges corresponding to the correlation coefficients cut off values are not correctly defined. Examine the literature better and determine the intervals correctly.

5) Discussions

Problems with translation, already described in the result, have been repeated. Remove it from one of the two parts.

A lot of information given in the result has been repeated in the discussion. In the discussion, features differentiated from previous studies should be highlighted.

Reviewer #2: Abstract: The following sentence is unclear. «Known-groups comparison showed that the

questionnaire discriminated well for those who differed in education (p < 0.05) but age

(p > 0.05).». What is meant by global BBQ-score ? Does the global BBQ score include the distractor items ?

Introduction page 12: back pain altitudes should be changed into back pain attitudes (« and clinicians to identify back pain altitudes and beliefs and design appropriate interventions»)

Methods : What does conveniently recruited mean ? (« The participants were conveniently recruited into the study »). Please report the participation rates in four different hospitals. What about the treatment of patients ? Patients who received behavioral cognitive therapy should be excluded, because change of their BBQ is a goal in therapy.

It is unclear why the distractor items were included in some testings what author(s) label test of global BBQ score (e.g., « Known-groups validity: Known-groups validity was assessed by comparing the global and 9-item scores with age and education levels using one-way ANOVA.

Morover, in exploratory factor analysis, the distractor items were included, too. Inclusion of distractor items in factor analysis blurres the results. A confirmatory factor analysis that tests the 9 items of the BBQ is a better test of the on-factor-structure.

Results : Please report differences in BBQ mean levels between four samples. It is surprising that no missing values were observed. Do the author(s) have an explanation ?

Reviewer #3: General comment: very interesting paper, well written and well structured. The statistical analysis is varied, and results brings new data. The conclusion answers correctly the scientific question addressed here.

Comment 1: Abstract is too long to my opinion.

C2: Why do the authors claim that they need 100 subjects and have planned to recruit 200?

C3: The questionnaires were administered a second time at D7-D14. How can the authors guarantee that no intervention was done during this delay prone to change patients' health status?

C4: I suggest moving the table 1 to e-addenda as I guess all readers will not be interested by these technical considerations.

C5: Method: This section is difficult to follow. Please clarify. I would suggest reordering the undersections. For example, first a few lines on the questionnaire with the global method; second, the translation procedure; third, the population and the authorization; fourth, the validation with the design of the study and content of the validation (dimensions explored in your study); fifth, the stats. The section results is very easy to follow, a similar construction may help (to my opinion).

C6 Method: I suggest limiting the number of tables in the article and for example delete the table 1 and move the informations in the text in the method section or the discussion.

C7 Method section/stats/Point 4/ Line3: The authors talk about table 2 relating result from previous studies. Please check.

C8 The design is not clear for the validity and the reliability not presented as different in the method section and separated in the results. Please check. The method used needs to as clear as possible. Was it the same sample for both?

C9 Table 3: I suggest deleting the column Highest and lowest score, useless.

C10: Why Table 7 is before Table 5? Please check all the tables and reorder.

C11 Discussion: Item 4 is not related to wheelchair.

C12 Discussion, Paragraph 4: “In the present study, a low negative correlation between these measures was found contrary to the Norwegian adaptation [35] where a moderate correlation was established.” What does “contrary” mean here: opposite (positive correlation) or intensity (low vs moderate)? Please clarify

C13 Discussion Paragraph 6: Table 8 does not refer to the factor structure, please check.

C14 Discussion paragraph 6: “In a similar passion” does not look appropriate in a scientific report.

C15 Discussion paragraph 6: The authors found a one factor solution for the nine item scoring. Item 4 (only ceiling effect detected) loaded on the fourth factor. Item 8 (wheelchair) loaded on the third one. The internal consistency is not graphically represented, and I wonder if the item 8 is useful for interpretation (last items on the Cronbach scheme). A figure representing the explanation part per item in this scpecific case may likely be interesting.

C16 Discussion Last paragraph: “The major limitation is the lack of cause effects relationship of the intervention”. It is not a therapeutic intervention, so it is difficult to understand.

C17 Conclusion: Consider that the BBQ is not directly useful for prevention or reduction of LBP consequences but more for detection and categorization of specific consequences of low back pain.

C18 Table 6: please check the SD value of the 18-24 group.

C19: Congratulations to the authors, very nice work.

6. PLOS authors have the option to publish the peer review history of their article (what does this mean?). If published, this will include your full peer review and any attached files.

Reviewer #1: **Yes: **Okan Küçükakkaş

Reviewer #2: No

Reviewer #3: **Yes: **Arnaud Dupeyron

---

## [Author Response · Author response to Decision Letter 0]

29 Jan 2021

Response letter to the reviewer's comments for the manuscript (PONE-D-20-31974) submitted to PLOS ONE

Response to Reviewer #1 

Dear reviewer,

We would like to thank you for the time you spent reviewing our manuscript entitled “The Hausa back beliefs questionnaire: Translation, cross-cultural adaptation and psychometric assessment in mixed urban and rural Nigerian populations with chronic low back pain”. 

You can find all the modifications/changes [text highlighted in red (deletions) and purple (additions/corrections)] in the revised file (marked version).

Please find below our response to your constructive comments/suggestions.

Comment 1 (General):

Please review for spelling and grammatical mistakes.

Response/amendments:

Spelling and grammatical mistakes were reviewed and corrected throughout the manuscript. 

Comment 2 (Abstract): 

It's been too long. it should be summed up more briefly.

Response/amendments:

The abstract has been reduced as you suggested. 

Comment 3 (Introduction): 

a. The paragraph that starts as “Most developed patient-reported… “and gives general information about PROMs contains unnecessary information. 

b. This paragraph can be omitted.“Nigeria is Africa's most populous…” This paragraph as was also lengthened unnecessarily. In this paragraph, the necessity of the Hausa version should be stated more briefly.

Response/amendments:

a. The paragraph was omitted as you suggested. 

b. The paragraph modified/reduced 

Comment 4 (Methods): 

a. Cronbach’s and ICC values given for the original BBQ are not in excellent ranges. Look again, which value corresponds to which range.

b. No reference is given to the PCS Hausa version.

c. The first paragraph in “Assessment of outcomes” was also very unnecessary. We can take it out in this part.

d. The ranges corresponding to the correlation coefficients cut off values are not correctly defined. Examine the literature better and determine the intervals correctly.

Response/amendments:

a. The range of the accepted values of Cronbach’s α and ICC for the BBQ has been revised based on the range of values defined in the literature and those reported in similar validation studies (for a priori hypotheses). 

b. The reference to Hausa version of the PCS is now added. 

c. The first paragraph under “Assessment of outcomes” was modified/reduced rather than deleted because we think that is important to describe how the data of the questionnaires were collected. Different hospitals and different raters were used. Moreover, interviewer administration method was utilized as majority (60%) of the respondent were non-literates. 

d. The ranges of the correlation coefficients cut off values were revised and updated as you suggested. 

Comment 5 (Discussion):

a. Problems with translation, already described in the result, have been repeated. Remove it from one of the two parts.

b. A lot of information given in the result has been repeated in the discussion. In the discussion, features differentiated from previous studies should be highlighted.

Response/amendments:

a. Problems with translation were removed from the discussion as you suggested. 

b. Repeated information in the discussion were removed where necessary and features differentiated from previous studies were highlighted as suggested. 

Once again, we thank you for all your constructive suggestions/comments. Indeed, your review has significantly improved our manuscript. 

Note: Grammatical errors, typographic errors and technicalities were corrected throughout the manuscript. 

Thank you. 

Sincerely.

Response to Reviewer #2 

Dear reviewer,

We would like to thank you for the time you spent reviewing our manuscript entitled “The Hausa back beliefs questionnaire: Translation, cross-cultural adaptation and psychometric assessment in mixed urban and rural Nigerian populations with chronic low back pain”.

You can find all the modifications/changes [text highlighted in red (deletions) and purple (additions/corrections)] in the revised file (marked version).

Please find below our response to your constructive comments/suggestions.

Comment 1 (Abstract):

a. The following sentence is unclear. «Known-groups comparison showed that the questionnaire discriminated well for those who differed in education (p < 0.05) but age

(p > 0.05).». 

b. What is meant by global BBQ-score? Does the global BBQ score include the distractor items?

Response/amendments:

1. The statement “Known-groups comparison showed that the questionnaire discriminated well for those who differed in education (p < 0.05) but age (p > 0.05)” implies that the comparison of the patients’ BBQ scores according to their education status (subgroup) was statistically significant suggesting that the questionnaire may be able to differentiate patients’ pessimistic beliefs based on their education status. However, for the age variable, the comparison was not statistically significant suggesting that the questionnaire cannot differentiate patients’ pessimistic beliefs according to their age.

2. Yes, the global or total BBQ score means the scores for all the questionnaire items (14 items) including the distractors. 

Comment 2 (Introduction): 

Page 12: back pain altitudes should be changed into back pain attitudes (« and clinicians to identify back pain altitudes and beliefs and design appropriate interventions»)

Response/amendments:

Many thanks for your observation. It was out of sight and has been corrected. 

Comment 3 (Methodology): 

a. What does conveniently recruited mean? (« The participants were conveniently recruited into the study »).

b. Please report the participation rates in four different hospitals.

c. What about the treatment of patients? Patients who received behavioral cognitive therapy should be excluded, because change of their BBQ is a goal in therapy.

d. It is unclear why the distractor items were included in some testings what author(s) label test of global BBQ score (e.g., « Known-groups validity: Known-groups validity was assessed by comparing the global and 9-item scores with age and education levels using one-way ANOVA.

e. Moreover, in exploratory factor analysis, the distractor items were included, too. Inclusion of distractor items in factor analysis blurres the results. A confirmatory factor analysis that tests the 9 items of the BBQ is a better test of the on-factor-structure.

Response/amendments: 

a. We wanted to say “consecutively recruited” instead. This has been corrected. 

b. Sincerely, it will be difficult for us to report the precise participation rates in the four different hospitals as the questionnaires were mixed and we did not indicate the address or hospital name on each of the questionnaires. 

c. None of the patients received cognitive behavioral therapy (even though we did not mention this in the inclusion/exclusion criteria). However, we could not rule out that some patients received other treatments most especially electrotherapy or exercise. 

d. We have now excluded the global BBQ score from external construct validity (convergent, divergent, and known-groups validity) and reliability analyses (internal consistency, ICC, SEM, MDC95, and LOA95%) as inclusion of the distractors items is not relevant since they are not included in computing the BBQ scores.

e. Generally, when conducting EFA of a measure, all items are included to have an insight of the putative structure including the distractors’ items will verify the underlying structure of the BBQ. This was also done in all the adaptions that examined EFA of the BBQ (e.g. the Arabic, Turkish and Yoruba versions) However, when it comes to CFA, the distractors are not included since the purpose is to verify the underlying latent constructs (i.e. the 9 scoring items of the BBQ). 

Comment 4 (Results):

a. Please report differences in BBQ mean levels between four samples. 

b. It is surprising that no missing values were observed. Do the author(s) have an explanation?

Response/amendments: 

a. As we previously stated (response to point 3b above), it will be difficult for us to report the mean of the four samples separately.

b. Potential missing values were evaluated by the raters (physiotherapists) as most of the questionnaires were interviewer-administered. It was done by cross-checking all the items to ensure that respondents did not leave any item unanswered. The raters received training prior to the data collection and they were reminded that all questions should be checked for missing values.

Once again, we thank you for all your constructive suggestions/comments. Indeed, your review has significantly improved our manuscript. 

Note: Grammatical errors, typographic errors and technicalities were corrected throughout the manuscript. 

Thank you. 

Sincerely.

Response to Reviewer #3 

Dear reviewer,

We would like to thank you for the time you spent reviewing our manuscript entitled “The Hausa back beliefs questionnaire: Translation, cross-cultural adaptation and psychometric assessment in mixed urban and rural Nigerian populations with chronic low back pain”.

You can find all the modifications/changes [text highlighted in red (deletions) and purple (additions/corrections)] in the revised file (marked version).

Please find below our response to your constructive comments/suggestions

Comment 1: Abstract is too long to my opinion.

Response/amendments: The abstract is now reduced as you suggested.

Comment 2: Why do the authors claim that they need 100 subjects and have planned to recruit 200?

Response/amendments: Yes, 100 subjects is the minimum recommended by COSMIN and Terwee et al (2007) guidelines, however, the patients were recruited as part of the lead author PhD project to cross-culturally adapt and validate low back pain measures (e.g. ODI, RMDQ, SF-12. FABQ, NPRS, PCS e.t.c.) not only the BBQ, into Hausa. Adaptation of these measures requires adequate sample size especially when conducting Rasch analysis, EFA and CFA as large sample size is generally recommended to obtain robust and precise item parameter estimates.

Comment 3: The questionnaires were administered a second time at D7-D14. How can the authors guarantee that no intervention was done during this delay prone to change patients' health status?

Response/amendments: This is one of the limitations of our study and we have mentioned it in the discussion aspect. We cannot guarantee that no intervention was given during the test-retest period, however, due nature of clinic schedules in our environment, where patients are usually given weekly (once per week) or two-weekly (once per two weeks) appointment/follow-up, we, therefore, believe that even if the patients have received an intervention, it may not significantly affect the patients’ health status. 

Comment 4: I suggest moving the table 1 to e-addenda as I guess all readers will not be interested by these technical considerations

Response/amendments: We deleted Table 1 since it is not that important and we have many tables in the manuscript. 

Comment 5: Method: This section is difficult to follow. Please clarify. I would suggest reordering the under sections. For example, first a few lines on the questionnaire with the global method; second, the translation procedure; third, the population and the authorization; fourth, the validation with the design of the study and content of the validation (dimensions explored in your study); fifth, the stats. The section results is very easy to follow, a similar construction may help (to my opinion).

Response/amendments: The methods section is now modified for clarity. 

Comment 6: Method: I suggest limiting the number of tables in the article and for example delete the table 1 and move the information in the text in the method section or the discussion.

Response/amendments: Table 1 was deleted and the contents were moved in the text in the method section (statistical analyses) as you suggested. Reviewer #1 also suggested limiting the number of tables.

Comment 7: Method section/stats/Point 4/ Line 3: The authors talk about table 2 relating result from previous studies. Please check.

Response/amendments: It was out of sight. We have corrected this. 

Comment 8: The design is not clear for the validity and the reliability not presented as different in the method section and separated in the results. Please check. The method used needs to as clear as possible. Was it the same sample for both?

Response/amendments: The same sample was used for the validity and reliability. In the method section, we clearly explain this for clarity.

Comment 9: Table 3: I suggest deleting the column Highest and lowest score, useless. 

Response/amendments: Highest and lowest score columns in Table 3 were deleted as you suggested. 

Response/amendments: The same sample was used for the validity and reliability. In the method section, we clearly explain this for clarity.

Comment 10: Why Table 7 is before Table 5? Please check all the tables and reorder.

Response/amendments: All tables were checked and reordered as you suggested.

Comment 11: Discussion: Item 4 is not related to wheelchair.

Response/amendments: Thank you for observing this error. The sentence is now corrected. 

Comment 12: Discussion, Paragraph 4: “In the present study, a low negative correlation between these measures was found contrary to the Norwegian adaptation [35] where a moderate correlation was established.” What does “contrary” mean here: opposite (positive correlation) or intensity (low vs moderate)? Please clarify

Response/amendments: The paragraph has been corrected for clarity.

Comment 13: Discussion Paragraph 6: Table 8 does not refer to the factor structure, please check

Response/amendments: Table 8 is now Table 7. There is a column (last) for the factor structure of the BBQ assessed by previous studies, hence we cited the table. 

Comment 14: Discussion paragraph 6: “In a similar passion” does not look appropriate in a scientific report.

Response/amendments: The phrase “In a similar passion” was replaced with “in the same vein”

Comment 15: Discussion paragraph 6: The authors found a one-factor solution for the nine item scoring. Item 4 (only ceiling effect detected) loaded on the fourth factor. Item 8 (wheelchair) loaded on the third one. The internal consistency is not graphically represented, and I wonder if the item 8 is useful for interpretation (last items on the Cronbach’s scheme). A figure representing the explanation part per item in this specific case may likely be interesting.

Response/amendments: We agree that since item 8 did not load on the first factor (9 scoring item factor), it may has an ambiguous meaning to the population, hence, not measuring the same construct as the other eight items. Surprisingly, the retaining or removal of this item did not significantly changed the alpha coefficients (i.e. α = 0.75 vs. 0.78), hence this may justify retaining the item in BBQ scoring. For item 4, we are not so bothered since it belongs to the distractor items, and retaining or deleting the item did not significantly changed the alpha coefficients (i.e. 0.722 vs. 0.743) for the whole questionnaire (global score). We commented on this briefly in the discussion section. The ICC and other reliability as well as validity analyses for the whole questionnaire was deleted based on the recommendation of other reviewer that distractor items are not used in scoring the BBQ. 

Below is the internal consistency analysis output from SPSS (for 9-item score)

Item-Total Statistics

 Scale Mean if Item Deleted Scale Variance if Item Deleted Corrected Item-Total Correlation Cronbach's Alpha if Item Deleted

BBQ-Q1 20.59 23.842 .561 .765

BBQ-Q2 20.48 23.155 .340 .806

BBQ-Q3 20.56 22.137 .684 .746

BBQ-Q6 20.73 24.681 .503 .773

BBQ-Q8 20.53 25.456 .313 .796

BBQ-Q10 20.57 23.844 .507 .771

BBQ-Q12 20.84 24.390 .449 .778

BBQ-Q13 20.97 24.818 .474 .776

BBQ-Q14 20.71 22.508 .638 .752

Below is the internal consistency analysis output from SPSS (for global score)

Item-Total Statistics

 Scale Mean if Item Deleted Scale Variance if Item Deleted Corrected Item-Total Correlation Cronbach's Alpha if Item Deleted

BBQ-Q1 33.405 46.323 .438 .698

BBQ-Q2 33.295 44.480 .333 .708

BBQ-Q3 33.380 44.518 .523 .687

BBQ-Q4 33.005 49.030 .077 .743

BBQ-Q5 33.550 48.982 .058 .750

BBQ-Q6 33.550 46.701 .445 .699

BBQ-Q7 33.645 44.160 .458 .692

BBQ-Q8 33.350 47.957 .258 .715

BBQ-Q9 33.770 44.047 .448 .693

BBQ-Q10 33.390 46.018 .422 .698

BBQ-Q11 33.535 44.632 .317 .711

BBQ-Q12 33.655 46.458 .391 .702

BBQ-Q13 33.785 47.124 .397 .703

BBQ-Q14 33.530 45.175 .468 .693

Comment 16: Discussion Last paragraph: “The major limitation is the lack of cause effects relationship of the intervention”. It is not a therapeutic intervention, so it is difficult to understand.

Response/amendments: We deleted the statement regarding such limitation since it is understood that our study is not interventional as you suggested. 

Comment 17: Conclusion: Consider that the BBQ is not directly useful for prevention or reduction of LBP consequences but more for detection and categorization of specific consequences of low back pain.

Response/amendments: Thank you for the constructive suggestion. We modified the conclusion while taking note of the BBQ is rather used for detection and categorization of specific consequences of low back pain as you suggested.

Comment 18: Table 6: please check the SD value of the 18-24 group.

Response/amendments: It was out of sight and has been corrected. 

Comment 19: Congratulations to the authors, very nice work.

Response: Once again, we thank you for all your constructive suggestions and comments. Indeed, your review has significantly improved our manuscript. 

Note: Grammatical errors, typographic errors and technicalities were corrected throughout the manuscript. 

Thank you. 

Sincerely.

---

## [Decision Letter · Decision Letter 1]

2 Mar 2021

PONE-D-20-31974R1

The Hausa Back Beliefs Questionnaire: Translation, cross-cultural adaptation and psychometric assessment in mixed urban and rural Nigerian populations with chronic low back pain

PLOS ONE

Dear Dr. Ibrahim,

Thank you for submitting your manuscript to PLOS ONE. After careful consideration, we feel that it has merit but does not fully meet PLOS ONE’s publication criteria as it currently stands. Therefore, we invite you to submit a revised version of the manuscript that addresses the points raised during the review process.

We look forward to receiving your revised manuscript.

Kind regards,

Adewale L. Oyeyemi, Ph.D

Academic Editor

PLOS ONE

Journal Requirements:

Reviewers' comments:

Reviewer's Responses to Questions

**Comments to the Author**

1. If the authors have adequately addressed your comments raised in a previous round of review and you feel that this manuscript is now acceptable for publication, you may indicate that here to bypass the “Comments to the Author” section, enter your conflict of interest statement in the “Confidential to Editor” section, and submit your "Accept" recommendation.

Reviewer #1: All comments have been addressed

Reviewer #2: All comments have been addressed

Reviewer #3: All comments have been addressed

2. Is the manuscript technically sound, and do the data support the conclusions?

Reviewer #1: Yes

Reviewer #2: Yes

Reviewer #3: Yes

3. Has the statistical analysis been performed appropriately and rigorously? 

Reviewer #1: Yes

Reviewer #2: Yes

Reviewer #3: Yes

4. Have the authors made all data underlying the findings in their manuscript fully available?

Reviewer #1: Yes

Reviewer #2: Yes

Reviewer #3: No

5. Is the manuscript presented in an intelligible fashion and written in standard English?

Reviewer #1: Yes

Reviewer #2: Yes

Reviewer #3: Yes

6. Review Comments to the Author

Reviewer #1: I reviewed the article with the changes made. Necessary corrections have been made. The article can now be accepted.

Reviewer #2: Author(s) should add to their limitations can not identify the four hospitals in questionnaires and check for potential bias. Author8s) should also add to their limitations that the data collection was interview-based. Interview-based data collection might overestimate the clearness and readability of the questionnaire.

Reviewer #3: I would like to thank the authors for their extensive effort to improve the text. They have correctly answered my questions and comments.

Some minor comments for the revised manuscript:

1. Once again, the lines are not numbered and it is difficult to localize the comments

2 Translation and cross-cultural adaptation section: I suggest to give the name of the "translators" (initials if authors or full names)

3 Translation and cross-cultural adaptation section, last sentence: The authors claim having sent the final version to the original developers and what happened?

4 I Would suggest to replace "sample size estimation" by "population" and move the two last sentences (patients needed for reliability and validity before statistical analysis) in this section.

5 In the result section does not need to report the results in the text and in the table (external validity for example) please choose one or the other.

6 Discussion, first paragraph: for Item 4 a ceiling effect would likely mean that patients believe that doctors are unable to help them. Please check.

7 The discussion is interesting.

7. PLOS authors have the option to publish the peer review history of their article (what does this mean?). If published, this will include your full peer review and any attached files.

Reviewer #1: **Yes: **Okan Lüçükakkaş

Reviewer #2: No

Reviewer #3: **Yes: **Arnaud Dupeyron

---

## [Author Response · Author response to Decision Letter 1]

6 Mar 2021

Response to Reviewer #1 

Dear reviewer,

Once again, we would like to thank you for the effort and time you spent reviewing our manuscript.

Sincerely.

Response to Reviewer #2 

Dear reviewer,

We would like to thank you for the effort and time you spent reviewing our manuscript for the second time. 

1. We have now added the response rate of the questionnaire according to the four hospitals (SEE General aspect, ceiling and floor effects, in the results section) as you suggested.

2. The limitation regarding interviewer-administration method was added in the discussion section as suggested. 

You can find all the modifications/changes [text highlighted in red (deletions) and purple (additions/corrections)] in the revised file (marked version).

Sincerely.

Response to Reviewer #3 

Dear reviewer,

We appreciate the effort and time you spent reviewing our manuscript for the second time. 

You can find all the modifications/changes [text highlighted in red (deletions) and purple (additions/corrections)] in the revised file (marked version).

Please find below our response to your constructive comments/suggestions

Comment 1: Once again, the lines are not numbered and it is difficult to localize the comments

Response/amendments: We apologize for this. The numbering is supposed to be handled by the journal electronic submission process so that reviewers may be able to see the lines numbered for easy reference. However, we have added the numbering in the revised files.

Comment 2: Translation and cross-cultural adaptation section: I suggest to give the name of the "translators" (initials if authors or full names)

Response/amendments: We have now provided the initials of the forward and backward translators. 

Comment 3: Translation and cross-cultural adaptation section, last sentence: The authors claim having sent the final version to the original developers and what happened?

Response/amendments: The final version of the Hausa-BBQ along with the report of the translation process was sent to the original developers for appraisal. The developers responded and there was no need for further modifications. We have now indicated this in the manuscript (SEE item 6: Proofreading).

Comment 4: I would suggest to replace "sample size estimation" by "population" and move the two last sentences (patients needed for reliability and validity before statistical analysis) in this section

Response/amendments: We replaced “Sample size estimation” with “Population” as you suggested. 

Comment 5: In the result section, does not need to report the results in the text and in the table (external validity for example) please choose one or the other. 

Response/amendments: The modifications were done. Table 5 was deleted and the order of the tables was reflected. 

Comment 6: Discussion, first paragraph: for Item 4 a ceiling effect would likely mean that patients believe that doctors are unable to help them. Please check

Response/amendments: The sentence was checked but ceiling effects means higher scores (disagree, as the scores are reversed) and this indicate that most patients had higher scores for item 4 (Doctors cannot do anything for back trouble), [mean score (SD) = 3.06 (1.36) as shown in Table 2]. Since the scores are reversed, the interpretation will be that most of the participants disagree or do not believe that doctors cannot do anything for back trouble.

Comment 7: The discussion is interesting.

Response: Once again, we thank you for all your constructive suggestions and comments. 

Sincerely.

---

## [Decision Letter · Decision Letter 2]

17 Mar 2021

The Hausa Back Beliefs Questionnaire: Translation, cross-cultural adaptation and psychometric assessment in mixed urban and rural Nigerian populations with chronic low back pain

PONE-D-20-31974R2

Dear Dr. Ibrahim,

We’re pleased to inform you that your manuscript has been judged scientifically suitable for publication and will be formally accepted for publication once it meets all outstanding technical requirements.

Kind regards,

Adewale L. Oyeyemi, Ph.D

Academic Editor

PLOS ONE

Additional Editor Comments (optional):

A very good manuscript that would make important contribution to the field. I am wondering if the authors can consider the inclusion of their questionnaire as a supplement to the manuscript.

Reviewers' comments:

Reviewer's Responses to Questions

**Comments to the Author**

1. If the authors have adequately addressed your comments raised in a previous round of review and you feel that this manuscript is now acceptable for publication, you may indicate that here to bypass the “Comments to the Author” section, enter your conflict of interest statement in the “Confidential to Editor” section, and submit your "Accept" recommendation.

Reviewer #2: All comments have been addressed

Reviewer #3: All comments have been addressed

2. Is the manuscript technically sound, and do the data support the conclusions?

Reviewer #2: Yes

Reviewer #3: Yes

3. Has the statistical analysis been performed appropriately and rigorously? 

Reviewer #2: Yes

Reviewer #3: Yes

4. Have the authors made all data underlying the findings in their manuscript fully available?

Reviewer #2: Yes

Reviewer #3: Yes

5. Is the manuscript presented in an intelligible fashion and written in standard English?

Reviewer #2: Yes

Reviewer #3: Yes

6. Review Comments to the Author

Reviewer #2: The author(s) addressed all my comments successfully. The ms improved considerably. Author(s) might think of adding their questionnaire as supplement to their paper.

Reviewer #3: No more comment. Just a suggestion in the method section: In the population paragraph delete "participants" which is covered by "population"

7. PLOS authors have the option to publish the peer review history of their article (what does this mean?). If published, this will include your full peer review and any attached files.

Reviewer #2: No

Reviewer #3: **Yes: **Arnaud DUPEYRON

---

## [Editor Report · Acceptance letter]

5 Apr 2021

PONE-D-20-31974R2 

The Hausa Back Beliefs Questionnaire: Translation, cross-cultural adaptation and psychometric assessment in mixed urban and rural Nigerian populations with chronic low back pain 

Dear Dr. Ibrahim:

I'm pleased to inform you that your manuscript has been deemed suitable for publication in PLOS ONE. Congratulations! Your manuscript is now with our production department. 

Kind regards, 

on behalf of

Dr. Adewale L. Oyeyemi 

Academic Editor

PLOS ONE